

# A within-subject comparison of different relaxation therapies in eliciting physiological and psychological changes in young women

Sarah Dib,  Jonathan C.K. Wells and  Mary Fewtrell

UCL Great Ormond Street Institute of Child Health, University College London, London, United Kingdom

## ABSTRACT

**Background**. Stress reactivity can be different in women compared to men, which might consequently influence disease risk. Stress in women may also generate adverse physiological effects on their offspring during pregnancy or lactation. The objective of this study was to compare the effects of different relaxation interventions on physiological outcomes and perceived relaxation in healthy young women, to assist in identifying the most appropriate intervention(s) for use in a subsequent trial for mothers who deliver prematurely.

**Methods**. A within-subject study was conducted in 17 women of reproductive age comparing five different relaxation interventions (guided-imagery meditation audio (GIM), music listening (ML), relaxation lighting (RL), GIM+RL, ML+RL), with control (silence/sitting), assigned in random order over a 3–6 week period. Subjective feelings of relaxation (10-point scale), heart rate (HR), systolic and diastolic blood pressure (SBP, DBP), and fingertip temperature (FT) were measured before and after each technique

**Results**. All interventions significantly increased perceived relaxation and FT, while music also significantly reduced SBP ($p < 0.05$). Compared to control, HR significantly decreased following GIM (mean difference = 3.2 bpm, $p < 0.05$), and FT increased (mean difference = 2.2 °C, $p < 0.05$) and SBP decreased (mean difference = 3.3 mmHg, $p < 0.01$) following ML. GIM + RL followed by GIM were the most preferred interventions.

**Conclusions**. Based on preference, simplicity, and the physiological and psychological effects, GIM and ML were identified as the most effective tools for reducing stress and improving relaxation. These techniques warrant further research in larger samples and other populations.

# INTRODUCTION

The body responds to stressors, whether physiological or psychological, by mounting a "fight or flight" response characterized by increased sympathetic nervous system activation. This results in increased blood pressure (BP), heart rate (HR) and glucocorticoid production (*Brotman, Golden & Wittstein, 2007*). The stress response is assumed to have evolved to

Corresponding author
Sarah Dib, sarah.dib.15@ucl.ac.uk

ensure the survival of the organism, and as the two sexes may face different sources of stress, the nature of the response may itself differ between males and females (*Kajantie & Phillips, 2006*). However, chronic stress or an exaggerated response can be harmful (*McEwen, 2008*), and again could differ by gender.

Moreover, stress in women of reproductive age may have consequences not only for the individual, but potentially for her offspring. Maternal stress during pregnancy is associated with preterm delivery, low birth weight, and even adverse infant cognitive development (*Beydoun & Saftlas, 2008*; *Bhang et al., 2016*; *Rondó et al., 2003*; *Talge, Neal & Glover, 2007*). Similarly, during lactation, symptoms of maternal distress correlate negatively with breastfeeding success (*Lau, 2001*), and potentially with infant growth and behaviour (*Mohd Shukri et al., 2019*). Interventions to reduce maternal stress thus have the potential to improve mother and infant outcomes, and to lower the risk of pregnancy and breastfeeding complications. However, the use of psychotherapy in this population is challenging due to stigma and financial and logistical barriers (*Goodman, 2009*). This suggests a need for alternative simple therapies.

Relaxation techniques such as music listening (*Knight & Rickard, 2001*), relaxation training (*Bastani et al., 2005*) and mindfulness (*Vieten & Astin, 2008*) are reported to reduce stress and increase relaxation. Some studies have also described a reduction in stress hormones, such as cortisol (*Chellew et al., 2015*; *Dolbier & Rush, 2012*), and in symptoms of anxiety and depression (*Manzoni et al., 2008*). Light therapy might also reduce stress by stimulating the suprachiasmatic nucleus in the hypothalamus and modulating the release of cortisol and adrenocorticotrophic hormone (*Pail et al., 2011*). Studies have shown a reduction in HR, BP, oxygen consumption and carbon dioxide production following exposure to blue light (*Cajochen et al., 2005*; *Litscher et al., 2013*). Bright Light Therapy, which utilises blue light of high intensity (∼7000–10000 lux), has been consistently found in several RCTs to influence mood disorders such as seasonal affective disorder (*Lam et al., 2006*; *Strong et al., 2009*) and non-seasonal depression (*Lam et al., 2016*; *Wirz-Justice et al., 2011*). Colourful ambient lighting has also been used in medical settings and was shown to improve patient satisfaction and lower pain ratings (*Robinson & Green, 2015*).

Different stress markers have been used to measure the stress response. Psychometric tools, such as the Visual Analogue Scale (VAS) for Stress or Perceived Stress Scale, were developed and widely used by the medical community to assess cognitive changes due to stress. However, these tools are subjective, and it is not known whether they correlate with physiological markers of stress, especially acute stress. Moreover, VAS has shown good within-subject reliability and validity, but not between participants (*Stubbs et al., 2000*). Objective markers that assess the hormonal response to stress are also used; cortisol being the most commonly measured. The disadvantages of using these markers include the relative invasiveness, cost of analysis and difficulty of performing measurements in daily life. Other objective markers of stress include physiological measurements such as heart rate, blood pressure, skin conductance, skin temperature and respiratory rate. Currently, there is no general consensus on the optimal way to measure stress. Therefore, in this study, we selected the most feasible, non-invasive physiological and psychological markers which could be used in daily life to assess stress/relaxation.
It is unknown whether relaxation techniques differ in their ability to reduce stress measured by physiological and psychological parameters. Moreover, most studies test the effectiveness of relaxation techniques in populations diagnosed with mood disorders, and it is unclear whether similar physiological changes are observed in healthy or at risk individuals. Lastly, the relation between perceived relaxation/stress and physical relaxation measured objectively is unclear.

In this study we compared the effects of five simple relaxation interventions on physiological markers and perception of stress in women of reproductive age. Our aim was to establish the most effective technique(s) for use in this population, but also to identify the most promising interventions that are feasible for use in a future study investigating stress reduction in women who deliver prematurely.

## MATERIALS & METHODS

### Study design

This was a within-subject study, where each participant received five different relaxation techniques and one silence/sitting control state in random order over a 3-6 week period, to minimize carry over effects (Fig. 1). The primary outcomes –perceived relaxation (PR), HR, BP and fingertip temperature (FT) - were measured at baseline prior to each intervention and post-intervention. The participants were reimbursed at the end of the study for their time. The protocol and procedures were reviewed and ethically approved by the UCL Research Ethics Committee (ref 12521/001) and the trial was registered at Clinicaltrials.gov (NCT03592147).

Participants were recruited by advertisement using flyers displayed at University College London, UK. Interested women contacted the researchers and received information about the study. After checking eligibility and obtaining written informed consent, the participants were enrolled in the study and assigned a random order of interventions generated using a computerised random number generator.

Each participant was allocated one or two time slots per week (based on their preference and availability) to undergo the five relaxation interventions (guided-imagery meditation (GIM) audio, music listening (ML), "relaxation" lighting (RL), combination of RL and GIM, and a combination of RL and ML) and control sessions. All sessions took place in a private, quiet room with controlled lighting and room temperature. To reduce variability due to circadian rhythm effects on hormones and mood, all sessions were scheduled between 9AM and 1PM, whenever possible. The participants were asked to leave their belongings including any electronic devices or books outside the room

Baseline information was collected, including date of birth, ethnicity, occupation, history of treatment of psychiatric illness, and if currently pregnant or breastfeeding. The Perceived Stress Scale (PSS), a 10-item instrument that measures the extent to which one's life is perceived as stressful or uncontrollable and overwhelming over the last month (*Cohen, Kamarck & Mermelstein, 1983*), was used to assess stress levels at baseline and at the end of the study.
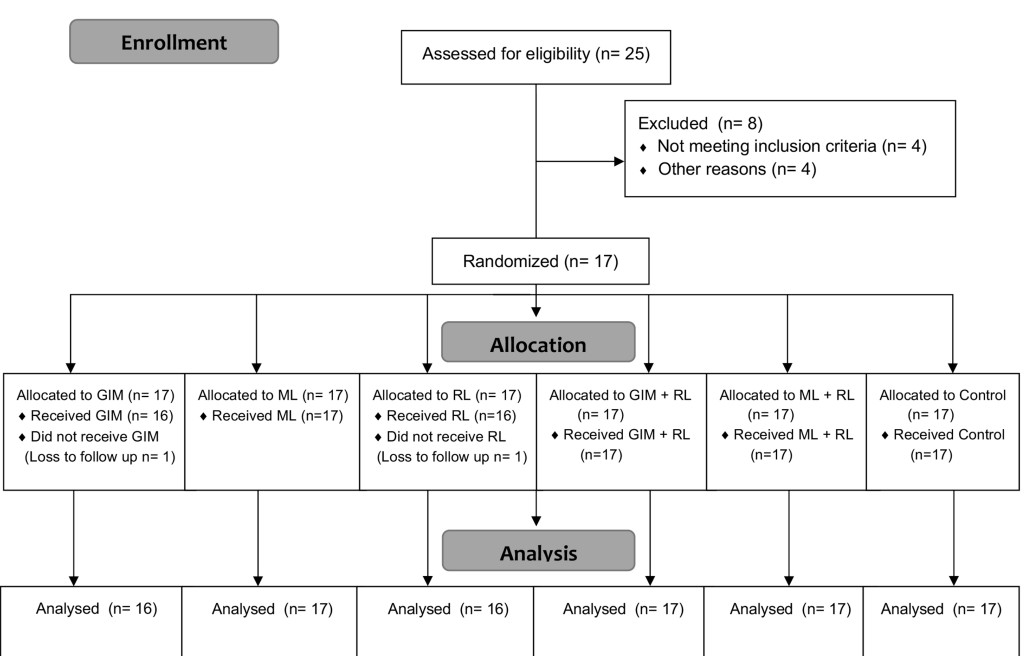

**Figure 1 CONSORT Flow Diagram for this within-subject trial.** Diagram shows participant flow through each stage of the trial (enrolment, allocation and analysis). One participant was unable to attend all the sessions and thus did not receive GIM nor RL. GIM: Guided-Imagery Meditation, ML: Music Listening, RL: Relaxation Lighting.

## Eligibility Criteria for Participants

Adult women of reproductive age (18-45 years), who were fluent in English, were eligible to participate. The exclusion criteria included smokers, having any recent surgery or injury, and/or having any condition that may affect BP, HR, or energy expenditure, such as hypertension or hypothyroidism.

## Interventions
### Guided Imagery Meditation Tape

Each participant was played, using headphones, a modified version of a GIM tape (*Menelli, 2004*). It has been previously successfully used in three randomized controlled trials for breastfeeding mothers (*Feher et al., 1989*; *Keith, Weaver & Vogel, 2012*; *Mohd Shukri et al., 2019*). The modified version excluded parts pertinent to breastfeeding to adapt it to our sample and was approximately 7 min in duration.

### Music Listening

Research suggests that to enhance the effects of music for relaxation, the music must be individualized based on the preferences of the participants. Therefore, we offered the women the option of selecting one of the following music categories: New age/guitar solo ("Internal Flight"), classical ("Pavane pour une Infante Défunte in G Major M.19"; "Gnossienne no 5"), and oriental ("Way to Heaven"; "Dune"). The songs were selected based on criteria established in a previous study to induce relaxation, including slow tempo

below or at resting HR (<72 bpm), repetitive rhythm, predictable dynamics, pleasant harmonies, and tonal qualities that include strings, flute or piano (*Knight & Rickard, 2001*; *Robb, 2000*). All songs were also modified in length to be approximately 7 min in duration and played through headphones.

### Relaxation Lighting

In a study in Singapore, cool blue light had the most relaxing effect, according to patient and staff feedback, whereas in Germany, warm orange light was preferred (Philips & I Timmers, pers. comm., 2017). Therefore, we asked the participants to select either the orange ("Relax" setting) or blue ("Energise" setting) lighting using the Philips Hue White and Colour Ambiance system (Philips, Netherlands). The intensity of the light was fixed to control for that measure. RL also lasted for 7 min in duration.

*Combination Interventions (Meditation + Relaxation Lighting; Music Listening + Relaxation Lighting):*

GIM and ML were combined with RL to investigate whether they had a synergistic effect on relaxation.

*Control/Silence:*

The participants were asked to relax for a duration of 7 min, with no explicit advice given. Lighting was adjusted to a specific intensity and colour (basic yellow light) as was used in the ML and GIM interventions.

### Outcome measures

The primary outcomes of this study were the mean changes in PR, HR, systolic blood pressure (SBP), diastolic blood pressure (DBP) and FT. PR was assessed using a visual analogue scale, which is a 10 cm horizontal line spanning from the minimum to the maximum of the variable measured. The minimum (left) represents "completely unrelaxed" and the maximum (right) "completely relaxed". The women marked a point on the scale to indicate their feelings of relaxation prior to and after each treatment. The distance between the mark and the minimum point was measured in centimetres (two decimal points) and the difference between pre- and post-intervention values were analysed. After the participant rested for 5 min, HR and BP were also measured pre- and post-relaxation technique using a digital sphygmomanometer (Omron M6, Japan). Lastly, a non-contact digital thermometer (Omron Gentle Temp 720, Japan) was used to measure FT as an indication of sympathetic nervous system activation. These measurements were taken three times and the mean was used for analysis. At the end of the study, the women were asked to rank the relaxation interventions in order of preference, and were given the option to make comments about the different techniques.

### Sample size

In a previous study investigating the effect of audio-visual imagery on patient anxiety and physiological measurements, including HR, a sample of 51 patients was able to demonstrate a significant reduction in HR (mean change (Z)= −0.75, SD = 1.00; $p = 0.01$) (*Vogel et al., 2012*). Using a within-subject design, a sample of 15 participants would be able to detect a 0.75 mean difference in HR following relaxation significant at 5% with a power of 80%.

*Statistical analysis*

Data were analysed using SPSS 21.0. First, the pre- and pos $t$-test changes were assessed using a paired $t$-test to test the effectiveness of each intervention and control. Then, to compare the interventions to control, the change in each primary outcome from pre- to post-intervention was compared to the change from pre- to post-control using a paired $t$-test. Variations in baseline measurements prior to each intervention may influence the observed response to each intervention. To account for this, we calculated the unstandardized residuals and repeated the same analysis. Lastly, associations between the physiological (BP, HR, FT) and psychological (PR, perceived stress) measurements were explored using Pearson's and Spearman's correlation tests, as appropriate. *P*-values <0.05 were considered statistically significant.

# RESULTS

## Descriptive data

The descriptive characteristics of the participants are presented in Table 1. Seventeen women with mean age 30.3 years (±6.2) participated. Nine participants reported regularly using a form of relaxation such as meditation, music, and yoga, while the eight others did not seek such techniques. At baseline, the average score on the PSS was 15.8 ± 5.2. The majority (71%) of the participants experienced moderate stress levels (14–26 points) and the rest low stress (Table 1). There were no significant differences in physiological parameters (HR, BP, FT) or PR at baseline between low and moderately stressed participants ($p > 0.05$; File S1).

## Intervention results

### Changes in outcomes during interventions

Pre to pos $t$-test changes in PR, SBP, DBP, HR and FT were compared to test the effectiveness of each intervention and control. As shown in Table 2, PR increased significantly following all interventions and control. All interventions significantly increased FT but this was not seen during the control session. Music was the only intervention to produce significant reductions in SBP ($-4.1 \pm 5.0$ mmHg).

### Comparison with Control

Mean changes in PR, SBP, DBP, HR, and FT following each intervention were compared with control to ascertain the relative effectiveness of the interventions (Fig. 2). Compared with control, HR significantly decreased with GIM (mean difference = 3.2 bpm, $p = 0.015$). All interventions reduced SBP and increased FT; however, compared to control, the change was only significant with the music intervention (mean difference = $-3.3$ mmHg, $p = 0.049$ and 2.2 °C, $p < 0.01$, respectively). Change in PR (Fig. 3) was significantly higher than control following music, GIM, GIM + RL, and ML+RL ($p < 0.05$). Mean (SD) and effect sizes are presented in Table 3.

| Table 1 Descriptive characteristics of the sample ($N = 17$). | |
|---|---|
| **Characteristics** | ***n* (%)** |
| Age (years) | |
| 18–24 | 3 (17.6) |
| 25–29 | 5 (29.4) |
| 30–34 | 4 (23.5) |
| 34–39 | 3 (17.6) |
| 40–45 | 2 (11.8) |
| Ethnicity | |
| White | 9 (52.9) |
| Asian | 5 (29.4) |
| Arab | 2 (11.8) |
| Other | 1 (5.9) |
| Employment Status | |
| Full-time | 8 (47.1) |
| Part-time | 2 (11.8) |
| Student | 6 (35.3) |
| Unemployed | 1 (5.9) |
| Perceived Stress (Baseline) | |
| Low Stress | 5 (29.4%) |
| Moderate Stress | 12 (70.6%) |

### *Comparison with control (controlling for baseline measurements)*

Using the unstandardized residuals, the previous results were unchanged; compared to control, HR decreased significantly more following GIM, while SBP was reduced and FT increased significantly more following ML.

## Preferences

Forty-three percent of the participants selected GIM + RL as their most preferred intervention, followed by GIM (24.1%) then ML (14.3%) and ML + RL (14.3%).

### *Perceived stress*

At the end of the study, average PSS score ($15.9 \pm 5.6$) was not significantly different from PSS at baseline ($15.8 \pm 5.2$).

### *Perceived vs physiological relaxation/stress*

No association was found between change (from pre to post intervention) in PR and change in SBP, DBP, HR, and FT. However, in the cases where change in FT was greater than 1.5 °C, increases in PR (measured using VAS) and FT were significantly correlated ($r = 0.364$, p<0.05 and when >2 °C, $r = 0.470$, $p < 0.01$).

Higher perceived stress, measured by PSS at baseline, was associated with a lower reduction in SBP following music ($r = 0.615$, $p = 0.009$) and GIM + RL ($r = 0.495$, $p = 0.043$). Change in SBP in participants with low vs high PSS was $-7.9 \pm 4.1$ mmHg and $-2.5 \pm 4.7$ mmHg following music and $-4.2 \pm 3.1$ and $-0.8 \pm 5.9$ following GIM + RL. This indicates that subjects experiencing lower stress levels in the previous

Dib et al. (2020), *PeerJ*, DOI 10.7717/peerj.9217

**Table 2  Mean changes in perceived relaxation, blood pressure, heart rate and fingertip temperature during interventions.**

| Intervention | Perceived relaxation (cm) Mean (SD) | Difference (95% CI) | Systolic blood pressure (mmHg) Mean (SD) | Difference (95% CI) | Diastolic blood pressure (mmHg) Mean (SD) | Difference (95% CI) | Heart rate (bpm) Mean (SD) | Difference (95% CI) | Fingertip temperature (°C) Mean (SD) | Difference (95% CI) |
|---|---|---|---|---|---|---|---|---|---|---|
| GIM | | 2.54 (1.92 to 3.15)** | | −2.59 (−6.08 to 0.89) | | −1.91 (−4.61 to 0.79) | | −1.70 (−4.04 to 0.64) | | 1.80 (0.13 to 3.47)* |
| Pre | 4.6 (1.9) | | 110.8 (8.7) | | 70.5 (8.8) | | 70.7 (9.8) | | 28.4 (4.4) | |
| Post | 7.1 (1.9) | | 108.2 (8.3) | | 68.6 (8.1) | | 69.0 (10.8) | | 30.2 (3.9) | |
| Music | | 2.34 (1.71 to 2.97)** | | −4.10 (−6.69 to −1.50)** | | −0.98 (−3.54 to 1.57) | | −0.89 (−3.31 to 1.53) | | 3.46 (1.68 to 5.24)** |
| Pre | 5.7 (1.5) | | 111.4 (8.9) | | 71.0 (9.5) | | 70.5 (11.0) | | 27.0 (3.7) | |
| Post | 8.0 (1.3) | | 107.3 (9.1) | | 70.0 (8.3) | | 69.6 (10.0) | | 30.5 (3.7) | |
| RL | | 1.08 (0.54 to 1.61)** | | −3.10 (−6.40 to 0.19) | | 2.68 (−5.54 to 0.19) | | 0.20 (−2.00 to 2.39) | | 2.31 (1.00 to 3.62)** |
| Pre | 6.0 (1.3) | | 111.3 (8.1) | | 73.2 (7.7) | | 70.7 (10.1) | | 28.1 (4.6) | |
| Post | 7.0 (1.7) | | 108.2 (9.2) | | 70.5 (8.5) | | 70.9 (9.7) | | 30.4 (3.6) | |
| GIM + RL | | 2.11 (1.50 to 2.72)** | | −1.76 (−4.53 to 1.00) | | −1.00 (−3.65 to 1.65) | | −1.25 (−3.25 to 0.74) | | 2.45 (0.51 to 4.39)* |
| Pre | 5.7 (1.9) | | 109.8 (8.5) | | 70.0 (9.6) | | 71.6 (9.8) | | 28.8 (4.7) | |
| Post | 7.8 (1.3) | | 108.1 (8.5) | | 69.0 (8.0) | | 70.3 (9.1) | | 31.2 (4.1) | |
| Music + RL | | 2.04 (1.11 to 2.96)** | | −1.41 (−4.48 to 1.65) | | 0.60 (−1.73 to 2.92) | | −1.22 (−3.51 to 1.08) | | 2.37 (0.73 to 4.01)** |
| Pre | 5.1 (1.9) | | 111.8 (10.1) | | 71.3 (10.3) | | 73.1 (10.1) | | 27.8 (4.6) | |
| Post | 7.1 (1.6) | | 110.4 (9.7) | | 71.9 (8.7) | | 71.9 (9.4) | | 30.2 (3.9) | |
| Control | | 1.02 (0.51 to 1.52)** | | −0.84 (−3.07 to 1.38) | | −1.64 (−4.77 to 1.49) | | 1.11 (−0.77 to 2.99) | | 1.27 (−0.16 to 2.71) |
| Pre | 5.8 (1.3) | | 110.2 (8.9) | | 73.1 (9.4) | | 67.0 (10.3) | | 28.2 (4.3) | |
| Post | 6.8 (1.4) | | 109.3 (9.7) | | 71.5 (11.5) | | 68.1 (9.7) | | 29.5 (3.9) | |

**Notes.**

*$p < 0.05$.

**$p < 0.01$.

GIM, Guided-imagery Meditation; RL, Relaxation Lighting.
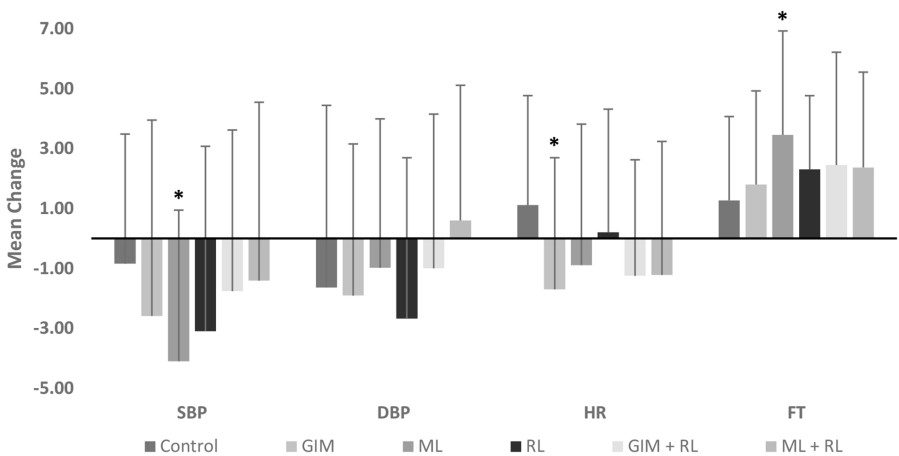

**Figure 2 Mean changes in physiological relaxation measures: Systolic blood pressure (SBP; mmHg), diastolic blood pressure (DBP; mmHg), heart rate (HR; bpm), and fingertip temperature (FT; °C).** Those significantly ($p < 0.05$) different from control were marked in asterisk (*). Bars indicate SD values. GIM: Guided-Imagery Meditation, ML: Music Listening, RL: Relaxation Lighting.

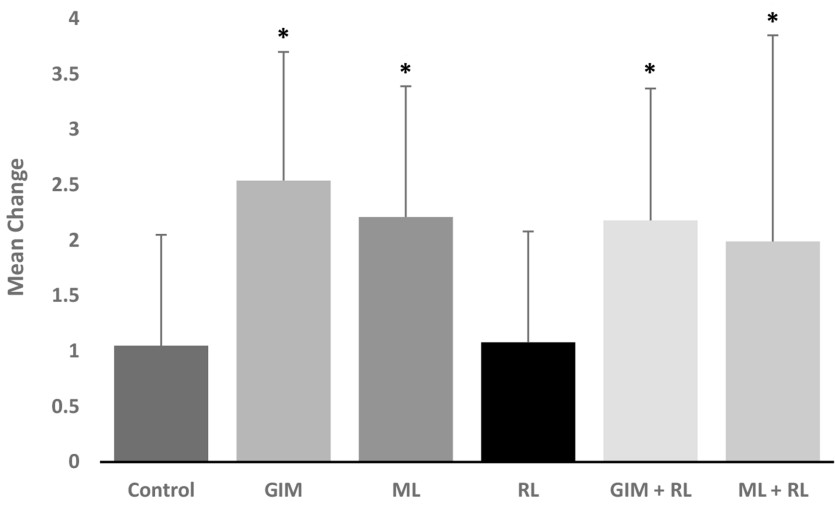

**Figure 3 Mean changes in perceived relaxation (psychological measure) measured on a 10-point visual analogue scale.** Those significantly ($p < 0.05$) different from control were marked in asterisk (*). Bars indicate SD values. GIM: Guided-Imagery Meditation, ML: Music Listening, RL: Relaxation Lighting.

month had greater reductions in SBP following relaxation interventions. Similarly, a lower reduction in SBP following music ($r = 0.556$, $p = 0.02$) and a lower increase in FT following silence/control ($r = -0.643$, $p = 0.005$) was associated with higher perceived stress at the end of the 3-6 week study. Participants with low vs high PSS at the end of the study had a mean change in SBP equivalent to $-6.8 \pm 5.8$ mmHg and $-2.2 \pm 3.6$ mmHg following ML, and a mean change in FT equal to $3.4 \pm 2.8$ °C and $-0.2 \pm 1.7$ °C following control/silence.

**Table 3    Mean differences in primary outcomes following each intervention compared to the control State.**

| Intervention | Perceived relaxation (cm) | | Systolic blood pressure (mmHg) | | Diastolic blood pressure (mmHg) | | Heart rate (bpm) | | Fingertip temperature (°C) | |
|---|---|---|---|---|---|---|---|---|---|---|
| | Mean (SD) | Effect size (d) | Mean (SD) | Effect size (d) | Mean (SD) | Effect size (d) | Mean (SD) | Effect size (d) | Mean (SD) | Effect size (d) |
| GIM | | | | | | | | | | |
| Control | 1.49 (1.09) | 1.4** | −1.78 (9.03) | −0.2 | −0.52 (8.19) | −0.06 | −3.15 (4.57) | −0.7* | 0.50 (2.84) | 0.2 |
| Music | | | | | | | | | | |
| Control | 1.26 (1.23) | 1.0** | −3.26 (6.31) | −0.5* | 0.66 (6.94) | 0.1 | −2.00 (6.23) | −0.3 | 2.19 (3.06) | 0.7** |
| RL | | | | | | | | | | |
| Control | 0.03 (1.27) | 0.02 | −2.29 (6.00) | −0.4 | −1.29 (8.25) | −0.2 | −1.25 (5.82) | −0.2 | 1.01 (3.90) | 0.3 |
| GIM + RL | | | | | | | | | | |
| Control | 1.09 (1.53) | 0.7** | −0.92 (5.31) | −0.2 | 0.64 (9.71) | 0.1 | −2.36 (5.88) | −0.4 | 1.18 (3.05) | 0.4 |
| Music + RL | | | | | | | | | | |
| Control | 1.02 (1.85) | 0.6* | −0.57 (8.13) | −0.1 | 2.24 (8.12) | 0.3 | −2.32 (6.42) | −0.4 | 1.10 (4.22) | 0.3 |

**Notes.**
*$p < 0.05$.
**$p < 0.01$.

## DISCUSSION

Previous studies have investigated light therapies, music listening, or meditation individually in relation to subjective assessment of anxiety, stress or depression. Only a few studied the effects of relaxation techniques on measurable physiological changes such as HR, BP, and FT. Moreover, many previous studies have focused on the short-term effects of stress interventions in special populations. Our study differs from previous studies as it assesses the short-term effects of different relaxation interventions in healthy participants.

All interventions and control significantly increased PR. Moreover, all interventions, but not the control, significantly increased FT while music significantly reduced SBP. When compared to control, however, GIM was the most effective intervention for increasing subjective feeling of relaxation and reducing HR. ML also significantly increased PR and FT, and decreased SBP compared to control.

The results are consistent with previous studies that showed any relaxation technique is capable of producing significant relaxation responses (*Regehr, Glancy & Pitts, 2013*). However, the advantage of the current study is the within-subject design that includes a control setting, which limits subject and environment variations. Comparing each intervention to the control demonstrated that GIM was the most effective overall when considering the physiological and psychological relaxation effects and preferences. ML produced a significant increase in FT, an indication of sympathetic nervous system stimulation, and a significant decrease in SBP. It is important to note that, on average, participants had lower baseline temperatures prior to the ML intervention. Having lower FT at baseline was associated with greater increases, whereas participants with higher FT at baseline had less potential for significant change. Thus, we controlled for the differences in baseline measurements by calculating the unstudentized residuals; however, the results remained unchanged indicating that music had an effect on FT and SBP regardless of the baseline value.

A similar within-subject study comparing the five interventions was recently conducted in Beijing, China, in mothers who were breastfeeding their healthy term infant (*Yu et al., 2018*). GIM was identified as the most effective relaxation intervention as it significantly reduced HR and increased PR compared to control, as seen in our study. Moreover, all treatments significantly increased PR. However, GIM also significantly decreased BP, and all treatments produced significant increases in FT and some reductions in BP, which was not observed in our study. The discrepancy in results might be due to the difference in characteristics of the two populations. Breastfeeding is associated with increased parasympathetic nervous system modulation and an attenuated stress response. Lactating mothers may also experience lower perceived stress levels, which might improve the response to relaxation interventions. The differences might also reflect cultural factors. The use of complementary and alternative medicine is more common in China than in the UK (*Dixon, 2008*), and is especially prevalent among lactating mothers (*Sim et al., 2013*). Breastfeeding participants in China might have been more receptive to the relaxation treatments than our sample of young women living in the UK, which might have enhanced their response to the interventions.

Interestingly, we found no association between the change in PR measured using VAS and the change in physiological measures (HR, BP, and FT). The discrepancy might be explained by the fact that subjective measurements such as the PSS and VAS are susceptible to recall bias, attention bias, and self-deception (*Goyal et al., 2016*). Conversely, objective measures rule out the possibility of these, but can be prone to measurement error and confounding effects of variables such as physical activity. The lack of association might be also due to the chemical basis by which the relaxation interventions were able to produce changes in physiological measurements. For example, oxytocin is known to produce anti-stress effects, which is evident in pregnant and lactating women who experience an attenuated stress response due to the increased levels of oxytocin (*Moberg & Prime, 2013*). Oxytocin has also been recognised as a cardiovascular hormone that can reduce BP and slow HR (*Gutkowska, Jankowski & Antunes-Rodrigues, 2014*). A recent study looking at the effect of relaxing (slow-tempo) and exciting (fast-tempo) music on salivary oxytocin and cortisol demonstrated that listening to relaxing music resulted in a significant increase in oxytocin (*Ooishi et al., 2017*). They also investigated the association between change in oxytocin and change in both HR and self-reported emotions (arousal). The ratio of increase in oxytocin was significantly related to the ratio of decrease in HR; however, oxytocin was not related to self-reported emotions. These results are consistent with our study, where the relaxation interventions elicited a physiological response which was not correlated with psychological/perceived relaxation. Further research is required to investigate the association between perceived, physiological, and biochemical mediators of relaxation.

By contrast, higher perceived stress measured by PSS was associated with lower changes in SBP and FT following certain interventions. Similar to our study, others have shown that the perception of stress is closely related to physiological measurements and might blunt cardiac reactivity to stressors (*Gartland et al., 2014*; *Ginty & Conklin, 2011*). Therefore, our results might indicate that, even though the participants experienced an increase in their PR measured by VAS following the different relaxation interventions, higher general perceived stress (measured using the PSS) might attenuate the more immediate physiological response to the interventions.

The majority of participants selected GIM + RL as their most preferred intervention, with the second most preferred options being GIM then ML. Physiologically, however, adding RL to GIM did not produce any additive effect, and it reduced the effect of GIM on HR and PR. GIM requires participants to follow verbal instructions, including concentrating on breathing and keeping eyes closed at certain time points. Conversely with the RL, participants might be inclined to focus on keeping their eyes open to benefit from the light. This might create sensory overload and counteract the relaxation response.

The strengths of the present study include the within-subject study design that also includes a control state. The main limitation of the study is the short duration of each intervention (6–7 min). Relaxation techniques such as lighting might require longer and repeated exposure to produce a relaxation response. However, we demonstrated that even with this short time period, some relaxation interventions could induce a response; we predict that prolonged and consistent use of those techniques might have further benefits. Further studies including a larger sample should investigate this. Another limitation

was that we did not collect information on the phase of menstrual cycle or hormonal contraceptive use, thus we could not assess its influence on the reactivity to relaxation interventions. Lastly, we did not adjust the $p$-value for multiple testing which should be considered when interpreting the results.

## CONCLUSIONS

In young women of reproductive age we were able to experimentally demonstrate that preference-based ML and GIM were the most effective interventions for reducing stress when compared to the control setting. These findings are informative in defining the most appropriate relaxation interventions for use in future studies aimed at stress reduction in women during pregnancy and lactation; by diverting the use of energy from stress and increasing energy investment in the infant, such interventions have the potential to improve pregnancy and breastfeeding outcomes. We found no association between perceived and physiological indicators of relaxation (HR, BP, and FT). Future studies should investigate this further, including biochemical indices of stress/relaxation, such as cortisol and oxytocin. Establishing associations between different indicators of stress/relaxation and clinical outcomes is also important in order to define the most appropriate outcome measures for use in studies testing relaxation interventions.

### Funding

This research is funded by a PhD studentship and funds from University College London Great Ormond Street Institute of Child Health. The funders had no role in study design, data collection and analysis, decision to publish, or preparation of the manuscript.

### Grant Disclosures

The following grant information was disclosed by the authors:
University College London Great Ormond Street Institute of Child Health.

### Competing Interests

The authors declare there are no competing interests.

### Author Contributions

- Sarah Dib conceived and designed the experiments, performed the experiments, analyzed the data, prepared figures and/or tables, authored or reviewed drafts of the paper, and approved the final draft.
- Jonathan C.K. Wells and Mary Fewtrell conceived and designed the experiments, authored or reviewed drafts of the paper, and approved the final draft.

### Human Ethics

The following information was supplied relating to ethical approvals (i.e., approving body and any reference numbers):

University College London Research Ethics Committee granted ethical approval to carry out the study within its facilities (Ethical Application Ref 12521/001).

## Data Availability

The raw measurements of blood pressure, heart rate, fingertip temperature, and perceived relaxation pre- and post-intervention are available in File S1.

## Supplemental Information

Supplemental information for this article can be found online at http://dx.doi.org/10.7717/peerj.9217#supplemental-information.

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
