# Peer review of "A within-subject comparison of different relaxation therapies in eliciting physiological and psychological changes in young women"

_PeerJ, doi:10.7717/peerj.9217_

## Round 0.1 · original submission · Minor Revisions

Three reviewers generally provided postive comments about your manuscript. Authors should pay special attention to amend in the manuscript the limitations highlighted by the reviewer 3.

Reviewer 1 ·

Basic reporting

I think this is a very interesting study, given the elegance of its design. It can be useful for clinical practice, both with healthy and ill populations.

However, I think the paper needs editing to be more understandable. Sentences should be shorter and more direct. Too many sentences with “and.” Results should be discussed more clearly.

Not trying to be flip, but I would just like to see tighter editing. In the Discussion section, too much is made of a Chinese study. I would rather see more discussion of the relationship (or lack thereof) of perceived relaxation and physiological measures.

Experimental design

Not sure how “Stairway to Heaven” (Led Zeppelin) can be characterized as “oriental” or meet the physiological criteria you state. However, it is a wonderful song.

Validity of the findings

Again, I think you Results & Discussion section need more editing. More direct discussion of each finding, eg, perceived relaxation and physio measures.

Additional comments

I really like your study in conceptualization, design and implementation. I feel it has implications for clinical practice, especially for high risk pregnancies. Would just like to see tighter writing with more concise organization.

·

Basic reporting

no comment

Experimental design

no comment

Validity of the findings

no comment

Additional comments

General comments
This manuscript aims at comparing the effects of five relaxation interventions on physiological markers and perception of stress in women of reproductive age. Authors manage to fulfill sufficiently their aim.

Minor comments
(lines 96÷98) Please, provide supporting reference;
(l446÷448) add reference details;
(l160) … Sheri Menelli in 2004. Provide a regular reference;
(l177) In a study in Singapore… Provide supporting reference;
(l180) … Philips Hue system. Add instrument’s details;
(Relaxation Lighting) Did Relaxation Lighting last 7’, as well?
(l414) add reference details.

·

Basic reporting

see below

Experimental design

see below

Validity of the findings

see below

Additional comments

This is an interesting trial that was carefully planned and well-conducted. I agree with the authors that the comparison of different relaxation interventions is indeed of great interest. A number of limitations apply, however, as outlined here:

1. The first two sections of the introduction seem unnecesassary. The authors probably want to justify their use of a female-only sample, but conceptually, some of the arguments that are made are not well-founded (e.g., the two sexes do not necessarily face different sources of stress; they face the same sources, but they might respond differently, which is something completely different). Even if it were the case, then one would still expect the authors to include a male sample just to test those hypothesized sex differences. The second section deals with the benefits of relaxation in the context of pregancy and breast-feeding, but since this was not a topic of the current study, I really don´t see how this might justify the use of a female-only sample. I agree that relaxation techniques are relevant in these conditions, but that´s something that should be part of the discussion, not the introduction.

2. In relation to issue number 1, the introduction should be used for providing a compelling rationale of the selection of the relaxation techniques (why these and why not others?) as well as the dependent variables. The latter is really important, because the reader is kind of suprised to learn about the main outcome variables only in the methods section; however, the authors need to introduce and justify their outcomes already in the introduction, based on sound theoretical reasoning.

3. The main limitation is the lack of control of reproductive variables. The authors should have, at least, assessed cycle phases and controlled for a potential effect on the main outcome variables. Also, it is unclear whether the use of hormonal contraceptives was exclusionary as there is evidence that HC might influence some of the outcome variables.

4. There is evidence that stress reactivity is lower in the morning compared to afternoon hours (e.g. cortisol). The time window from 9am to 1pm seems quite broad, particularly given the fact that there is a spike in cortisol at noon, and that having eaten lunch might affect relaxation parameters. It is doubtful, therefore, that participants examined in the morning can be compared to those examined at lunch hours. Also, I was wondering whether the same time was held constant within-person (e.g. the same person always coming to the lab at 9am).

5. Half of the participants „reported regularly using a form of relaxation“ (line 230). This is usually an exclusionary criterion, so I was wondering why the authors allowed for this variable. Did those individuals differ from the ones not practicing relaxation techniques?

6. The authors should adhere to standard CONSORT reporting for trials, as it is important to know the flow of recruiting, inclusion/exclusion of participants, drop-outs, etc.

7. A couple of smaller issues: a) the abstract states that all interventions increased PR, whereas in the discussion it is stated that all interventions, except RL and control, increased PR; the authors should check for inconsistencies between abstract and manuscript; b) the authors state (line 90f.) that the use of psychotherapy in this population is challenging, but do not provide a reason for this statement; c) throughout the manuscript, „interventions“ should be used instead of „therapies“; „relaxation techniques“ would be even better, as this adequately describes the interventions used in this study; d) the sample size is relatively small; an a priori power analysis is usually done for all main outcome variables, not just one (HR in this case); also, I was wondering about the (clinical) relevance of 0.75 mean difference in HR; the authors should better justify their power analysis – or call their study a „pilot study“; e) importantly, all analyses need to be controlled for multiple comparisons, resulting in p level adjustment; this might lead to some findings not being significant anymore; f) three out of four relaxation techniques were auditory-based; the authors should discuss whether this might affect the physiological pathways they are assuming, and whether and how it is ok to compare these techniques to light exposure which is something completely different.

---

## Round 0.2 · accepted · Accept

Reviewers are satisfied with the changes implemented in the manuscript. Congratulations!

Reviewer 1 ·

Basic reporting

No comment

Experimental design

No comment

Validity of the findings

No comment

Additional comments

I appreciate your making the changes I suggested. It is a better article, more able to be understood and implemented by other clinicians.

·

Basic reporting

no comment

Experimental design

no comment

Validity of the findings

no comment

Additional comments

General comments
I do not have any further particular concerns to express about the manuscript. Authors addressed sufficiently all points raised by the three reviewers.